# Can India's primary care facilities deliver? A cross-sectional assessment of the Indian public health system's capacity for basic delivery and newborn services

Jigyasa Sharma,[1] Hannah H Leslie,[1] Mathilda Regan,[1] Devaki Nambiar,[2] Margaret E Kruk[1]

[1]Department of Global Health and Population, Harvard TH Chan School of Public Health, Boston, Massachusetts, USA
[2]George Institute for Health, New Delhi, India

**Correspondence to**
Jigyasa Sharma;
jigyasa.sharma@mail.harvard.edu

## ABSTRACT

**Objectives** To assess input and process capacity for basic delivery and newborn (intrapartum care hereafter) care in the Indian public health system and to describe differences in facility capacity between rural and urban areas and across states.

**Design** Cross-sectional study.

**Setting** Data from the nationally representative 2012–2014 District Level Household and Facility Survey, which includes a census of community health centres (CHC) and sample of primary health centres (PHC) across 30 states and union territories in India.

**Participants** 8536 PHCs and 4810 CHCs.

**Outcome measures** We developed a summative index of 33 structural and process capacity items matching the Indian Public Health Standards for PHCs as a metric of minimum facility capacity for intrapartum care. We assessed differences in performance on this index across facility type and location.

**Results** About 30% of PHCs and 5% of CHCs reported not offering any intrapartum care. Among those offering services, volumes were low: median monthly delivery volume was 8 (IQR=13) in PHCs and 41 (IQR=73) in CHCs. Both PHCs and CHCs failed to meet the national standards for basic intrapartum care capacity. Mean facility capacity was low in PHCs in both urban (0.64) and rural (0.63) areas, while in CHCs, capacity was slightly higher in urban areas (0.77vs0.74). Gaps were most striking in availability of skilled human resources and emergency obstetric services. Poor capacity facilities were more concentrated in the more impoverished states, with 37% of districts from these states receiving scores in the lowest third of the facility capacity index (<0.70), compared with 21% of districts otherwise.

**Conclusions** Basic intrapartum care capacity in Indian public primary care facilities is weak in both rural and urban areas, especially lacking in the poorest states with worst health outcomes. Improving maternal and newborn health outcomes will require focused attention to quality measurement, accountability mechanisms and quality improvement. Policies to address deficits in skilled providers and emergency service availability are urgently required.

## Strengths and limitations of this study

► This study uses a nationally representative survey to assess primary public health facilities' adherence to nationally set minimum recommended standards for basic delivery and newborn (intrapartum) care.

► The index used to assess input and process capacity is based on national guidelines, and therefore represents a contextually appropriate measure of basic service readiness.

► However, the index lacked information on important technical and interpersonal care processes, such as provider competence, owing to data limitation.

► The sample did not include private facilities and included limited information on quality of referral care.

## INTRODUCTION

The adoption of Sustainable Development Goals (SDGs) has reaffirmed the reduction of preventable maternal and newborn deaths as global health priorities. Achieving the targeted reduction of the global maternal mortality ratio to less than 70 per 100 000 live births and neonatal mortality to 12 per 1000 live births[1] will require near universal coverage of institutional deliveries and timely detection and management of birth complications, as most maternal and newborn deaths occur at birth or within 24 hours of birth.[2] An emerging body of literature suggests that getting women to facilities alone will not suffice: what happens when a woman reaches a facility—including her right to quality antenatal, intrapartum and postpartum care— matters,[3 4] especially given the global mandate of universal health coverage.[5 6]

In India, increased coverage of facility-based births has not successfully translated to desired improvement in health outcomes for mothers and newborns. Under India's

National Rural Health Mission (NRHM, now called the National Health Mission or NHM), a variety of interventions were introduced through architectural improvements in the fund flow and design of services: increased number of maternal care facilities, particularly primary health centres (PHC) and community health centres (CHC); a strengthened supply chain for essential medicines, equipment and supplies; and Janani Suraksha Yojana, a financial incentive programme to increase institutional deliveries.[7] Following the launch of NRHM in 2005, institutional deliveries in rural areas have more than doubled,[8] and a declining trend in maternal and newborn mortality has been noted, although strong causal evidence linking NRHM efforts to improved health outcomes for mothers and newborns is lacking.[9–11] Annual decline in neonatal mortality between 2005 and 2015 was faster than in the preceding years; however, the rate of decline is not sufficient to meet the 2030 SDG targets.[12] Inadequate quality of care, including insufficient facility readiness, and low provider skill and clinical management capacity, as evidence from low/middle-income countries (LMIC) indicates, may explain why increased utilisation alone may not have resulted in the desired reduction in adverse intrapartum outcomes.[13 14] Moreover, quality of care itself also affects utilisation. Evidence from India indicates that the availability of a labour room and adequacy of essential equipment and laboratory services for childbirth at public health facilities have a significant effect on service uptake.[15 16] Describing and improving quality of intrapartum care is relevant to increasing service uptake in India, where maternal and newborn services are underused despite the availability of primary healthcare in public health facilities free of charge.

Public health facilities are a significant provider of care in India, especially for rural and vulnerable population segments. About 80.1% of all deliveries in rural India are facility based, of which about 70% are in public facilities.[17] For urban areas, 89.5% of births are institutional, 47.4% of which are in public facilities.[17] In the majority of India's states, more facility-based deliveries happen at government health institutions than in private facilities.[18 19] Additionally, quality of care in the public sector affects the poorest segment of the population the most, as the poorest wealth quintile is more heavily reliant on public health facilities than the richest, in both rural (58% vs 29%) and urban areas (48% vs 19%).[17 20]

India has made several efforts to strengthen its public health service provisioning. Initial policy emphasis was on rural areas given the preponderance of the Indian population living there and the assumption that population access to quality health services was relatively better in urban areas. However, mounting evidence of worse health outcomes for the urban poor compared with the urban rich—and sometimes even rural populations—led to the formal launch of the National Urban Health Mission in 2013.[21] The unified NHM is now charged with oversight for public healthcare in urban and rural areas.[22] The implementation of programmes and policies, however,

has varied subnationally. Under the NRHM, substantial funding and technical support has been provided to states with relatively weaker public health indicators and health system infrastructure, including the Empowered Action Group (EAG) states (Bihar, Jharkhand, Chhattisgarh, Madhya Pradesh, Uttar Pradesh, Uttaranchal, Orissa and Rajasthan).[23] About 46% of India's population resides in the EAG states, and this region lags behind the rest of the Indian states on socioeconomic, demographic and health indicators.[23]

To boost the performance of public health facilities, in 2007 the government introduced the Indian Public Health Standards (IPHS), a set of recommendatory standards to be used as a reference for public health infrastructure planning and as a benchmark for assessing the functional status of health facilities.[24] Although widely used in health system planning and management, the use of these standards in health system research has been limited in number and geographic scope to date. Rural health statistics from 2015 note that only 21% of PHCs and 26% of CHCs were functioning in concordance with IPHS standards, although a service-specific breakdown was not provided.[25] A few studies have used IPHS as a reference, noting deficiencies in service availability, human resource and infrastructure.[26–29] With one exception,[29] all these studies surveyed less than a hundred facilities.

The aim of this work is to understand the capacity of the Indian public health system to provide quality intrapartum care. This is an important element of understanding the effects of maternal and newborn health policies to date and setting priorities for health system strengthening going forward. In this paper, using the updated IPHS (2012) as the minimum standard for basic facility capacity for intrapartum care, we (1) assess the performance of PHCs and CHCs in India against this standard; and (2) describe differences in intrapartum care capacity between urban and rural areas, and across states.

## METHODS
### Study sample
We drew data from the fourth cycle of the District Level Household and Facility Survey (DLHS) conducted by the International Institute of Population Sciences in India. We used data from the facility modules of the DLHS for PHCs and CHCs to capture the primary level of the public health system designated to provide intrapartum care services. Typically, PHCs serve a population of 20 000–30 000,[30] and CHCs, which are intended to be referral centres for four PHCs, cater to approximately a population of 80 000–120 000.[30] PHCs are further categorized by delivery load (Type A and Type B); types are not distinguished in the DLHS and therefore could not be considered in this analysis. Within the tiers of the Indian public health system, CHCs are referral centres for PHCs; yet they are frequently used for first-line intrapartum care. In this paper, in keeping with

international guidelines, both are referred to as primary care facilities.[31]

The most recent survey, DLHS-4, was conducted in 2012–2014 and contains a sample of PHCs designed to be representative at district level and a census of CHCs. We also obtained the most recent district and state boundaries from the database of global administrative areas ( GADM.org) and matched DLHS data to the appropriate area using state and district name. We were unable to match three districts, likely due to district boundary changes between 2013 and 2016.

## Basic intrapartum care capacity index

Availability of essential physical and human resources and services is foundational to the provision of high-quality care.[32] The study objective was to evaluate basic intrapartum care capacity that should be present in all childbirth facilities. As described in the WHO quality of care framework for maternal and newborn health,[32] which builds on Donabedian's framework of healthcare quality,[33] structural capacity elements, such as availability of drugs and supplies, refer to necessary, but not sufficient, human and physical resource conditions for care delivery. Process capacity elements refer to facility's ability to offer evidence-based practices for routine care and management of complications, such as assisted vaginal delivery or administration of parenteral antibiotics.

We used the IPHS for PHCs to define intrapartum care capacity in this context; CHCs are assumed to meet at least the standards of the lower tier PHCs.[30] The facility capacity index comprises structure and process capacity elements required for both routine and basic emergency obstetric care.

We identified 48 elements of the IPHS pertaining to intrapartum care. Of these, we found at least some information on 33 in the DLHS facility audit, of which 26 were a complete match. Facility capacity to offer services such as manual removal of placenta, appropriate prereferral management of and referral for obstetric emergencies, and availability of select drugs and equipment, identified as the basic requirements for all PHCs and higher level facilities, were not included in the facility audit. See table 1 for the full list of IPHS requirements along with corresponding item from DLHS survey.

Our data comprised both observed and self-reported measures of structural inputs to care and self-reported process capacity indicators. Structural inputs fell into four subdomains: skilled provider availability (average of 4 indicators), facilities for a functional labour room (average of 9 indicators), emergency drugs and supplies for labour (average of 10 indicators), and delivery and newborn care equipment and supplies (average of 5 indicators). Process capacity indicators included reports from survey respondent (facility managers or medical officers in charge) on availability of assisted vaginal deliveries; administration of parenteral oxytocins, antibiotics, magnesium sulfate; postpartum haemorrhage management; newborn resuscitation and thermal protection services. All process capacity indicators were binary measures. The items were grouped following the IPHS categorisation. See online supplementary figure 1 for details on all items.

For both structure (four subdomains) and process (seven items) capacity indices, subdomains or indicators were averaged to provide a facility summary score from 0 to 1, with missing values excluded. Missing values were minimal (less than 3% of facilities) for all indicators except for availability of newborn resuscitation (11% of PHCs missing) and electricity supply with backup (23.6% of PHC and 9.6% of CHC missing). We created an index of facility capacity for basic intrapartum care as an average of the structure and process capacity indices. We followed current practice and weighted each indicator equally within a subdomain or index given the lack of guidance in the IPHS on weights.[34 35]

## Covariates

We identified covariates that may be associated with facility capacity. We followed DLHS classification of facilities as either a PHC or CHC, which is based on delivery volume and their capacity to perform comprehensive obstetric emergency management. We categorised facilities by rural versus urban location as reported in the survey. We also calculated annual delivery volume based on facility report, excluding missing values. We calculated capacity for quality intrapartum care within districts by weighting each facility by delivery volume. For weighting, we assigned facilities reporting childbirth capacity but missing delivery volume data or reporting no deliveries a delivery volume of 1.

## Statistical analysis

In this analysis, to describe primary level public health facilities' capacity to provide basic intrapartum care, we first calculated availability of basic childbirth care services at the facility level: facilities not offering any services, facilities offering round-the-clock services and those providing daytime services only. Some PHCs and CHCs may have curtailed services possibly due to their typology under IPHS, low volume (resulting from proximity to a higher level facility), inadequate infrastructure or human resources. As our data do not support classification of these facilities based on the reason for service unavailability (which may have been justified or not, given individual circumstances), we restricted subsequent analysis to facilities reporting provision of either round-the-clock or daytime only childbirth services. In this sample, we assessed each facility's adherence to national standard for care provision, stratified by facility type (PHC or CHC).

We described variation in facility-level capacity across rural and urban settings. We employed Pearson's $\chi^2$ test for binomial and Student's t-test for continuous indicators to assess the statistical significance ($p<0.05$) of difference in capacity between PHCs and CHCs and between rural and urban PHCs. To visualise differences in intrapartum care capacity across India, we also calculated averages of

**Table 1** Indian Public Health Standards for basic delivery and newborn care, and corresponding question (closest match) in the DLHS-4 facility survey for primary and community health centres

| Indian Public Health Standards | DLHS question | Data type | Categorisation |
|---|---|---|---|
| Management of normal deliveries | Can you tell me whether normal delivery services are provided in this facility? | Self-reported | – |
| Assisted vaginal deliveries including forceps/vacuum delivery whenever required | Can you tell me whether assisted (forceps delivery/vacuum) delivery services are provided in this facility? | Self-reported | Process |
| 24-hour delivery services both normal and assisted | Whether deliveries are conducted in this facility or no? If yes, whether the deliveries are conducted 24×7? | Self-reported | – |
| Manual removal of placenta | NA | | |
| Appropriate and prompt referral for cases needing specialist care | NA | | |
| Management of pregnancy-induced hypertension and referral | NA | | |
| Prereferral management (obstetric first aid) in obstetric emergencies that need expert assistance | NA | | |
| Proficient in identification and basic first aid treatment for PPH, eclampsia, sepsis and prompt referral | Can you tell me whether administration of parenteral antibiotics is provided in this facility? | Self-reported | Process |
| | Can you tell me whether administration of parenteral oxytocics is provided in this facility? | Self-reported | Process |
| | Can you tell me whether administration of parenteral magnesium sulfate is provided in this facility? | Self-reported | Process |
| | Can you tell me whether management of PPH is provided in this facility? | Self-reported | Process |
| Minimum 48 hours of stay after delivery | NA | | |
| Facilities for essential newborn care and resuscitation (newborn care corner in LR/OT) | Whether the following essential newborn care services are available? Resuscitation | Self-reported | Process |
| | Availability of designated newborn baby corner (functional) | Observed | – |
| Management of neonatal hypothermia, infection protection, cord care and identification of sick newborn and prompt referral | Whether the following essential newborn care services are available? Thermal protection (warmer/table lamp) | Self-reported | Process |
| Early initiation of breast feeding within 1 hour of birth | NA | | |
| At least one medical officer—MBBS | Availability of human resources (medical officer—regular)—No. In position | Self-reported | Structure: human resources |
| Three nurse-midwives (staff nurse) | Availability of human resources (ANM/female health worker—regular)—No. In position | Self-reported | Structure: human resources |
| Health worker (female) | NA | | |
| Health assistant (female) | NA | | |
| Training for staff for emergency management to be ensured | Training received by any medical officer during the last 5 years (SBA or Basic Emergency Obstetric Care training) | Self-reported | Structure: human resources |

Continued

**Table 1** Continued

| Indian Public Health Standards | DLHS question | Data type | Categorisation |
|---|---|---|---|
| | Training received for paramedical staff during the last 5 years (SBA training) | Self-reported | Structure: human resources |
| Labour table | Physically verify and record availability of labour table with McIntosh sheet | Observed | Structure: facilities and supplies for functional labour room |
| Managing labour using partograph | Partographs being recorded for the recently delivered women or women in labour at the facility | Self-reported | Structure: facilities and supplies for functional labour room |
| Suction machine | Physically verify and record availability of suction machine (functional) | Observed | Structure: facilities and supplies for functional labour room |
| Facility for oxygen administration | NA | | Structure: facilities and supplies for functional labour room |
| Sterilisation equipment (autoclave) | Physically verify and record availability of autoclave/steriliser (functional) | Observed | Structure: facilities and supplies for functional labour room |
| 24-hour running water | Physically verify and record availability of 24-hour running water supply | Observed | Structure: facilities and supplies for functional labour room |
| Electricity supply with backup facility (generator) | Whether the generator supply is connected to the LR | Observed | Structure: facilities and supplies for functional labour room |
| Attached toilet facilities | Physically verify and record availability of attached toilet in the LR (functional) | Observed | Structure: facilities and supplies for functional labour room |
| Delivery kits, including those for normal and assisted deliveries | Normal delivery kits available in the facility? | Self-reported | Structure: facilities and supplies for functional labour room |
| Privacy of woman ensured | Observe and record the condition of the LR—privacy in the LR (satisfactory) | Observed | Structure: facilities and supplies for functional labour room |
| Radiant warmer, fixed height with trolley | Radiant warmer | Observed | Structure: equipment and supplies for newborn care |
| Resuscitation bag and mask with reservoir | Ambu bag with mask (functional) | Observed | Structure: equipment and supplies for newborn care |
| Weighing scale | Baby weighing machine of any time (functional) | Observed | Structure: equipment and supplies for newborn care |
| Thermometer, clinical, digital | NA | | Structure: equipment and supplies for newborn care |
| Light examination | NA | | Structure: equipment and supplies for newborn care |
| Intravenous cannula 24G, 26G | Suction catheter/cannula (functional) | Observed | Structure: equipment and supplies for newborn care |
| Mucus extractor; pump suction, foot operated | Pedal suction machine/mucus extractor (functional) | Observed | Structure: equipment and supplies for newborn care |
| Feeding tube | NA | | |
| Oxygen catheter | NA | | |
| Sterile gloves | Whether following emergency drugs and consumables are available: at least two pairs of gloves | Self-reported | Structure: emergency drugs and supplies for labour and delivery |

Continued

**Table 1** Continued

| Indian Public Health Standards | DLHS question | Data type | Categorisation |
|---|---|---|---|
| Injectable oxytocin | Whether following emergency drugs and consumables are available: oxytocin injection | Self-reported | Structure: emergency drugs and supplies for labour and delivery |
| Injectable diazepam | Whether following emergency drugs and consumables are available: diazepam injection | Self-reported | Structure: emergency drugs and supplies for labour and delivery |
| Tablet nifedipine | Whether following emergency drugs and consumables are available: nifedipine tablet | Self-reported | Structure: emergency drugs and supplies for labour and delivery |
| Injectable magnesium sulfate | Whether following emergency drugs and consumables are available: magnesium sulfate injection | Self-reported | Structure: emergency drugs and supplies for labour and delivery |
| Injectable lignocaine hydrochloride | Whether following emergency drugs and consumables are available: lignocaine hydrochloride injection | Self-reported | Structure: emergency drugs and supplies for labour and delivery |
| Injectable methylergometrine maleate | NA | | Structure: emergency drugs and supplies for labour and delivery |
| Intravenous Haemaccel | NA | | Structure: emergency drugs and supplies for labour and delivery |
| Sterilised cotton and gauze | Whether following emergency drugs and consumables are available: sterilised cotton and gauze | Self-reported | Structure: emergency drugs and supplies for labour and delivery |
| Syringe | Whether following emergency drugs and consumables are available: sterile syringes and needles (different sizes) | Self-reported | Structure: emergency drugs and supplies for labour and delivery |
| Hub cutter | Hub cutter (available and functional) | Observed | Structure: emergency drugs and supplies for labour and delivery |
| Oxygen bottles | Whether following emergency drugs and consumables are available: oxygen cylinder with face mask, wrench and regulator—functional | Self-reported | Structure: emergency drugs and supplies for labour and delivery |

Indicators were extracted from Indian Public Health Standards (IPHS) for primary health centre; DLHS questions from both primary and community health centre surveys; NA refers to IPHS indicators that were not included in the DLHS facility survey.
ANM, Auxiliary Nurse Midwife; DLHS, District Level Household and Facility Survey; LR, labour room; OT, operation theatre; PPH, postpartum haemorrhage; SBA, skilled birth attendance.

overall facility intrapartum care capacity at the district level, weighting facilities by total deliveries. For mapping, we divided the overall facility capacity index scores into sixths, designating the lowest third of scores as 'poor capacity.'

Statistical analysis was done using Stata V.14.1 (StataCorp, Texas) and mapping using QGIS V.2.12 (Free Software Foundation, Massachusetts).

### Patient and public involvement
Patients or public were not involved in this analysis.

### RESULTS
The 2012–2013 DLHS-4 facility survey was conducted across India; however, data are publicly available only for 30 out of 36 states and union territories (excluding the states of Gujarat, and Jammu and Kashmir, and the territories Dadra and Nagar Haveli, Daman Diu, the National Capital Territory of Delhi, and Lakshadweep). Our study sample comprised a total of 8536 PHCs and 4810 CHCs. Although all PHCs and CHCs are expected to provide childbirth care, about 30.2% (n=2557) of PHCs and 5.2% (n=251) of CHCs reported not providing any childbirth services, either round the clock or daytime only. Only 59.7% (n=4798) of rural and 62.7% (n=312) of urban PHCs and only 92.7% (n=3578) of rural and 94.9% (n=895) of urban CHCs offered 24-hour intrapartum care. As figure 1 indicates, availability of services was similar for both levels of facility across urban and rural settings.

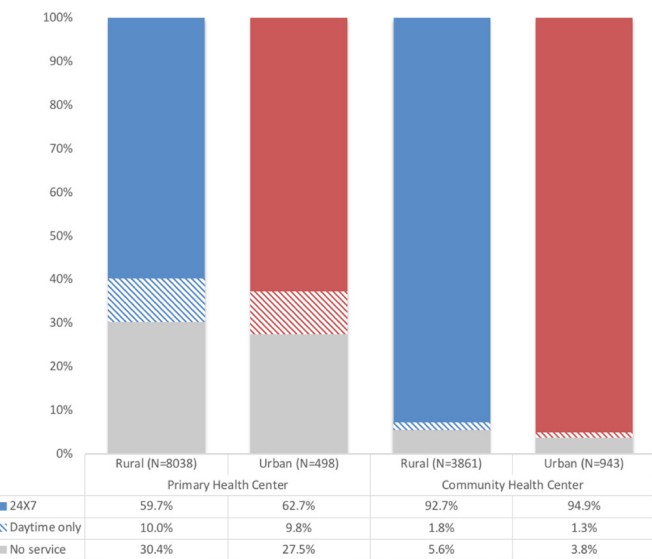

**Figure 1** Availability of basic delivery and newborn services in primary care facilities in India.

Within facilities offering any delivery services (n=5959 PHC, n=4553 CHC), median monthly delivery volume was 8 (IQR=13) in PHCs and 41 (IQR=73) in CHCs. All CHCs are expected to provide 24×7 delivery care; 73.4% of PHCs self-identified as government-designated 24×7 delivery facilities. In practice, 98% of CHCs and 86% of PHCs actually offered 24×7 services. A designated area for newborn care was present only in 67% and 80% of included PHCs and CHCs, respectively.

Overall capacity for basic intrapartum care was lower than the basic IPHS standard in both PHCs (mean 0.63, SD 0.23) and CHCs (mean 0.75, SD 0.17) included in this analysis. The structural capacity index for basic intrapartum care was slightly better than the process capacity index for both PHCs (mean 0.67, SD 0.19 vs 0.60, SD 0.29) and CHCs (mean 0.76, SD 0.14 vs 0.73, SD 0.22). As shown in figure 2, PHCs lagged behind CHCs on all indicators; the difference between PHCs and CHCs was statistically significant for all indicators and the summary

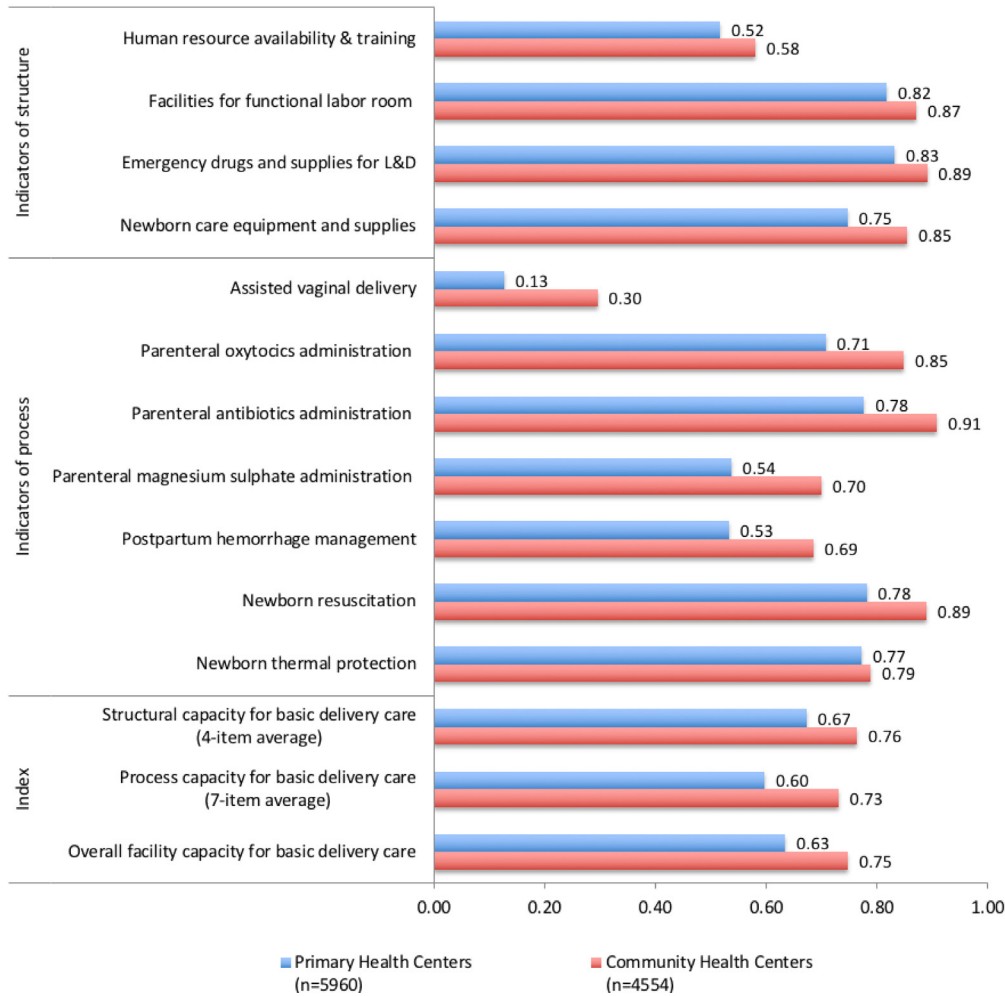

**Figure 2** Capacity of primary health centres (PHC) and community health centres (CHC) for basic delivery and newborn care. Overall facility capacity of basic delivery and newborn care calculated as the average of the two preceding summary measures (process and structural capacity). Details on indicators of structure are available in online supplementary figure 1. Differences between PHC and CHC for all indicators and summary index are statistically significant at p<0.05 level (Pearson's $X^2$ test and Student's t-test). L&D, Labour and Delivery.

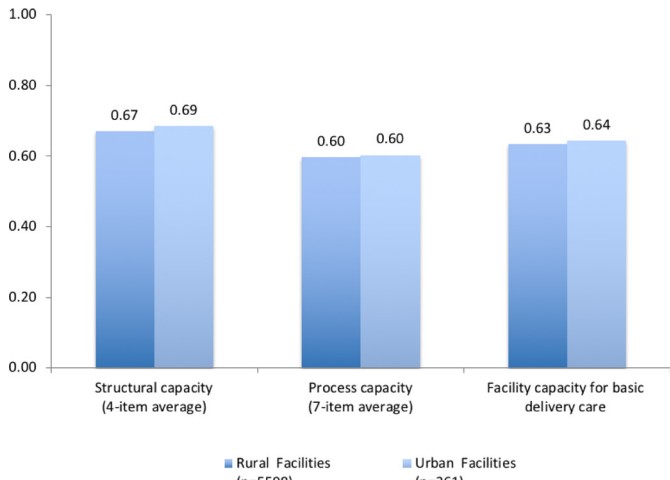

**Figure 3** Difference in facility capacity for basic delivery and newborn care across rural and urban primary health centres. Differences between rural and urban facilities for all three summary indices are not statistically significant at p<0.05 level (Student's t-test).

index. Although CHCs, as the referral centres for PHCs, are expected to offer more comprehensive care, they frequently lacked basic infrastructure required even at PHCs. Human resource availability and training in both CHCs (mean 0.58, SD 0.23) and PHCs (mean 0.52, SD 0.26) were comparably low. Particularly large gaps were seen in process capacity indicators such as provision of assisted vaginal deliveries (13% of PHCs vs 30% of CHCs), administration of parenteral magnesium sulfate (54% vs 70%) and management of postpartum haemorrhage (53% of PHCs vs 69% of CHCs).

Figures 3 and 4 display the differences in structural and process capacity indices, and in overall capacity of basic intrapartum care across rural and urban settings, for PHCs and CHCs, respectively. Facility capacity was low across the board, with small differences across geographical settings. Scores for structural, process and overall capacity summary indices were comparable

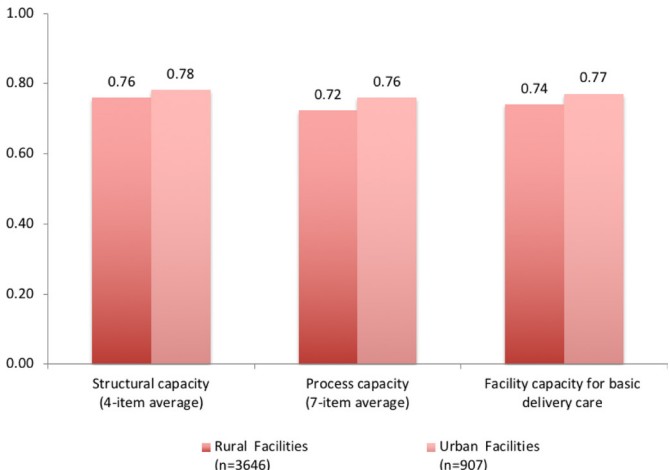

**Figure 4** Difference in facility capacity for basic delivery and newborn care across rural and urban community health centres.

for PHCs, with no significant rural-urban differences. Rural-urban differences in CHCs were slightly more pronounced, with urban CHCs outperforming rural CHCs by 0.02–0.04 in structural, process and overall facility capacity indices.

As shown in figure 5, the capacity of facilities that offered intrapartum care varied across the country, with districts with the poorest facility capacity concentrated in the northern part of the country. There were seven districts with no facilities offering childbirth services in the DLHS data. We had no data for 118 districts, including the districts in regions for which survey data were not publicly available. In both EAG and non-EAG states, there was significant variation in facility capacity (online supplementary figures 2 and 3). There was a disproportionate number of poor capacity facilities in EAG states, with 37% of districts scoring in the lowest third of the facility capacity index (less than 0.70) compared with 21% of non-EAG states. Among the nine EAG states, Uttar Pradesh had the greatest concentration of poor capacity facilities, with 84% of districts receiving scores in the lowest third.

## DISCUSSION

Using a national health facility survey, we found that many Indian public primary care facilities fail to meet the nationally set minimum recommended standards for basic intrapartum care. Despite a concerted effort to increase accessibility, about 30% of PHCs and 5% of CHCs did not offer any childbirth services; round-the-clock intrapartum care services were available only in 60% of PHCs and 94% of CHCs. Both PHCs and CHCs offering these services had critical deficiencies in routine and emergency care practices, infrastructure and staffing, fulfilling an average of only 63% and 75% of an index of basic intrapartum care capacity, respectively. These findings are consistent with studies documenting weak emergency care capacity, infrastructure and staffing in low-income countries, particularly in lower level facilities with low delivery volume.[34 36] Most CHCs, which are meant to serve as referral centres to manage more complicated cases,[23] failed to meet the threshold set for lower level facilities.

Structural capacity across the study facilities was marginally better than process capacity, with large gaps in provision of signal functions of emergency obstetric care.[37] For example, assisted vaginal deliveries were offered only in about one-tenth of PHCs and one-third of CHCs. Almost one-half of PHCs and one-third of CHCs reported not providing services to manage postpartum haemorrhage, a leading cause of maternal mortality in India. We also noted that availability of medicine and supplies, such as the partograph or uterotonics, was worse than that of equipment and infrastructure, such as scales or labour tables, underscoring the need to strengthen the supply chain for essential medicine and supplies. This is a challenge that the NRHM was designed to address, and one that has been resistant to change.[16]

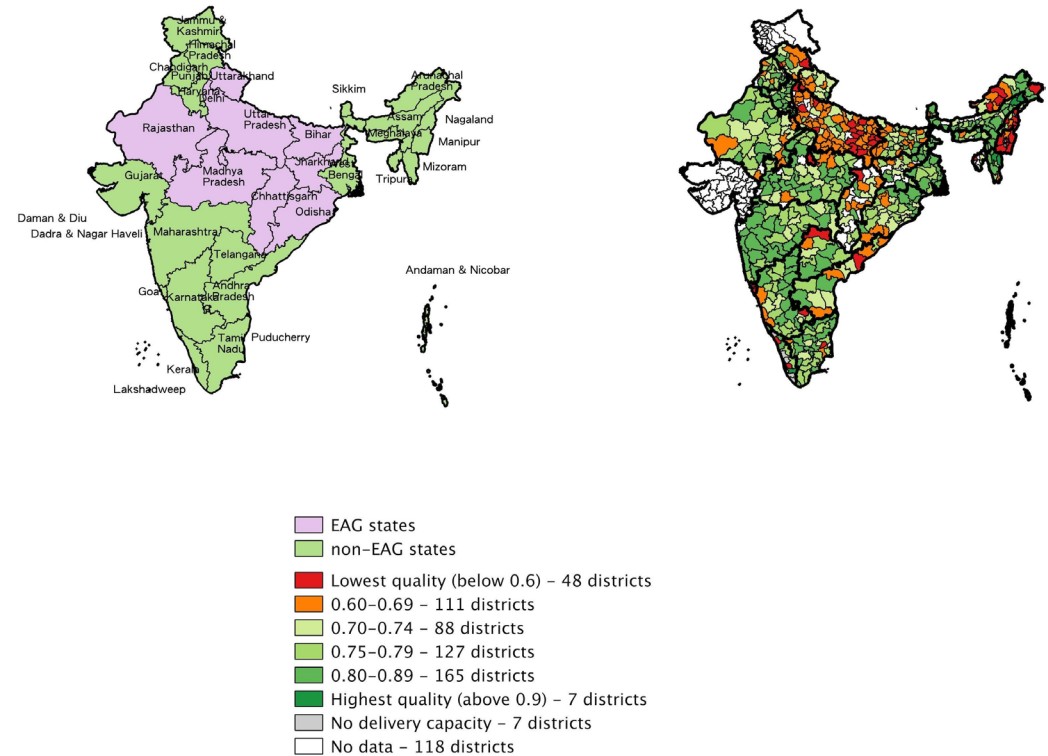

EAG states
non–EAG states
Lowest quality (below 0.6) – 48 districts
0.60–0.69 – 111 districts
0.70–0.74 – 88 districts
0.75–0.79 – 127 districts
0.80–0.89 – 165 districts
Highest quality (above 0.9) – 7 districts
No delivery capacity – 7 districts
No data – 118 districts

**Figure 5** Empowered Action Group (EAG) status and facility capacity for basic delivery and newborn care.

Of similar concern are the low scores on human resource availability and training across facilities. Although addressing the shortage of skilled providers in rural PHCs and CHCs has also been one of the agendas of the NRHM, critical deficiencies clearly persist, as has been documented elsewhere.[38] The gaps observed are especially problematic because our indicators reflect a minimum level of human resource preparedness (eg, skilled birth attendance training for auxiliary nurse-midwives or emergency obstetric care training for medical officers). They do not include key dimensions of provider availability and competence, such as appropriately diagnosing and managing complications, which the WHO identifies as integral to good quality of care during facility-based deliveries.[39] For example, accounting for absenteeism, a common problem in public facilities in India, would probably lower our estimate of actual human resource availability significantly.[38]

Facility capacity for basic intrapartum care in public facilities was similar in rural and urban areas. Our findings underline the need to improve facility capacity in both urban and rural areas, which the unified NHM may be well situated to tackle. Given the fact that the public facilities assessed here are the main source of obstetric care for poor populations in both rural and urban settings,[40 41] the deficiencies we found are likely to disproportionately affect the most vulnerable women. Meeting the commitment to equitable care inherent in the NHM requires redoubled attention on care quality in the public health system.

Despite additional funding and considerable technical support for improving maternal and newborn health

outcomes in the EAG states, facility capacity for intrapartum care remains weaker than in non-EAG states. We found that facilities with poor capacity were especially common in Uttar Pradesh and Bihar, states with the two highest maternal and newborn mortality rates in the country. Roughly comprising a third of India's population, these two states are critical in India's progress towards meeting national and international maternal and newborn mortality reduction targets.

Our findings indicate very low monthly delivery volume in PHCs. Bypassing of care at lower level facilities may be driven by myriad reasons,[42] including, but not limited to, women's preference for higher quality care when it is available, as seen in Tanzania.[43] Regardless of the reason, underutilisation of PHCs represents inefficiency in the Indian health system, considering the amount of resources spent at this level of healthcare.[44 22]

A demand for quality care, regardless of the level of facility, has been noted in India. Preference for higher level facilities with better quality is observed in the state of Kerala in India,[17] whereas in Tamil Nadu, women prefer to deliver in PHCs with qualified staff providing respectful care.[45] Most high-income and many middle-income countries encourage delivery in hospitals where surgical and newborn care services are available to all women.[46] Organising health system services around the capacity to deliver high-quality care may thus respond to women's observed preferences and improve health system performance. The breadth of quality deficits in CHCs and especially PHCs, which are under-resourced despite concerted efforts and investment of the NHM, is concerning. Rather than attempting to bring many

thousand facilities up to standards, regionalisation of obstetric care and quality improvement in high-volume facilities may offer a more efficient resource allocation strategy for the general population. Building on the success of NRHM in using supply-side and demand-side strategies to reduce barriers to facility-based delivery,[47] India may be particularly well positioned to assess the feasibility of regionalising care. Assessment of regionalisation should include checks against overmedicalisation of childbirth care, which has been seen in some states.[48] That said, it is essential to strengthen lower level facilities and improve referral linkages with higher level facilities in areas with highly remote, rural and/or vulnerable populations, to ensure continued access to services for all, especially considering evidence from India indicating preference for PHCs when they are adequate.[45 49] The current emphasis on upgrading service delivery at PHC and sub health centre level under Ayushman Bharat is an ambitious but critical first step in this direction.

Our study has several limitations. First, it would have been ideal to examine the quality of clinical care provided to women in public health facilities such as the use of basic emergency care procedures for mothers and newborns (signal functions) and adherence to global guidelines for routine deliveries.[35 50] Our index was constructed from indicators of basic equipment and service availability, measuring facility-level inputs to care and basic service readiness, but it lacked information on important technical and interpersonal care processes, such as provider competence. Previous studies have noted the importance of provider knowledge and effort in determining the quality of actual clinical encounters[51] as well as substantial variation in the skill and competence of skilled birth attendants.[52] Likewise, although disrespectful care during childbirth is common[53] and respectful care is identified as WHO as a key requirement for improving quality of intrapartum care,[39] we could not assess interpersonal care aspects in our index. Critically, data constrained our ability to assess private facilities, which are also relied on for intrapartum care and may have different adherence to IPHS standards. We were also not able to assess quality of referral care or the outcomes of care. Thus, our measure should be seen as a starting point in measuring care quality in India; process and outcome quality may well be worse than the largely input-based measures examined here. Finally, these findings may not be representative of the six states and union territories without the survey data.

Our study findings add to the growing body of literature documenting gaps in provision of quality intrapartum care in LMICs as well as inadequate measurement to assess health system quality. For 15 out of 48 intrapartum care items on the IPHS, we did not find any corresponding survey question in the DLHS facility audit. Harmonising data collection with national standards is a critical first step towards addressing information gaps. Likewise, expanding data collection to capture the nature and content of care being provided is important for understanding quality of care.[54] Quality of care is complex and multifaceted. While the inputs to care measured here are a critical foundation for high-quality care, they are a poor proxy of care as delivered.[55] Even measures of process quality may be insufficient to predict quality-sensitive outcomes such as morbidity and mortality,[10 56 57] particularly for childbirth where averting severe outcomes requires timely recognition of complications by qualified personnel and functional referral systems to ensure a rapid response.[10 57]

Future assessments should better capture the context of care and whether women's rights to quality care are being met, with a focus on clinical effectiveness of care, patient experience, respectful care and outcomes.[32] A variety of methods are available for this, including patient and provider interviews, direct observations and clinical vignettes; assessment of the private sector is essential to gaining a full understanding of health system performance.[51] Strengthening the publicly available health information system in India to collect reliable, frequent and timely data on facility quality is key for effective programme implementation and for meeting national and global targets such as those of the Ending Preventable Maternal Mortality, the Every Newborn Action Plan, and the SDGs. It is also essential to link quality measures to health outcomes and demographic characteristics of the population to assess the determinants of quality and its distribution across population subgroups. Health systems research in India needs to be expanded to address the large knowledge gaps in health system quality.[58]

## CONCLUSION

Primary care facilities in India are not well prepared to provide high-quality obstetric and newborn care, and facility capacity is worst in states with the worst health outcomes. Over the past decade, India's health system has operated in an extremely resource-constrained environment: from 2004 to 2014, government health expenditure has remained approximately around 1% of country's gross domestic product.[20] The Indian government will need to increase investment in the health system, in providers and in research to harness the full benefit of its public health infrastructure. Research on regionalisation is a priority as this may offer an innovative approach to ensuring quality services for mothers and newborns. The impact of regionalisation strategies on facility overcrowding, performance incentive structures for frontline workers, as well as equity in service access should be important considerations of such research. Improving quality of care and strengthening public health infrastructure is integral to India's path to universal health coverage: without an explicit focus on quality, a push towards universal coverage is unlikely to lead to better health for mothers and newborns.

**Contributors** JS, HHL and MEK conceived the research question. JS and HHL developed the methods, and JS, HHL and MR conducted the data analysis. JS, HHL, MR, DN and MEK interpreted the results, and JS wrote the first draft of the

manuscript. JS, HHL, MR, DN and MEK edited and revised the manuscript and approved the final draft for submission.

**Funding** This research received no specific grant from any funding agency in the public, commercial or not-for-profit sectors. DN is supported by a Wellcome Trust/Department of Biotechnology India Alliance Intermediate Investigator Award for Public Health.

**Competing interests** None declared.

**Patient consent** Not required.

**Ethics approval** The Harvard University Human Research Protection Program approved this analysis as exempt from human subjects review.

**Provenance and peer review** Not commissioned; externally peer reviewed.

**Data sharing statement** The data are publicly available through the International Institute of Population Sciences, India (website: http://www.iipsindia.ac.in).

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
