## [Reviewer comments · BMJ Open]

ARTICLE DETAILS

TITLE (PROVISIONAL)	Can India's primary care facilities deliver? A cross-sectional assessment of the Indian public health system's capacity for basic delivery and newborn services
AUTHORS	Sharma, Jigyasa; Leslie, Hannah; Regan, Mathilda; Nambiar, Devaki; Kruk, Margaret

VERSION 1 – REVIEW

REVIEWER	Meenakshi Gautham London School of Hygiene and Tropical Medicine, UK Based in India.
REVIEW RETURNED	17-Dec-2017

GENERAL COMMENTS	This is a very interesting paper and I am glad this analysis has been done. My major comments, listed below, are about the methods and the interpretation of results. My specific comments: Page 6, second para of the Introduction: The statements (lines 29-32 and 46-51) about JSY and delivery outcomes need to be presented with more factual accuracy and in a more nuanced way. Lim's study found an effect on perinatal and neonatal deaths which Powell Jackson's study did not. So one could consider this as mixed evidence but not as 'no improvement in delivery outcomes'. In terms of MMR, none of the studies found strong associations between JSY and maternal mortality but all authors have stated that insufficient study power and large standard errors could be one of the limitations. So, one could say that although maternal mortality has declined in India (from...to ...) during this period (as SRS data show), studies have not been able to establish a strong association between JSY and mortality reductions. You are also making a very strong case in the introduction, for quality being a weak link for maternal mortality improvements, and it's okay to emphasise this in the intro since you are building a rationale for the study objectives. However, in the discussion I feel you might need to raise some issues about the relationship between and mortality outcomes also, since a recent trial of a very big quality improvement programme in UP (Better Birth safety checklist) found No difference in neonatal or maternal outcomes (see Katherine Semrau et al, NEJM 2017). So there could be other factors like poor referral management for emergencies (as Powell-Jackson suggests) or perhaps something like deliveries being conducted by untrained workers even where trained staff are available (Gaurav Sharma et al, WHO Bulletin, 2016) etc, affecting mortality outcomes, besides quality. I would also argue for the importance of good quality care from a human rights and UHC perspective, and not just for mortality reductions.
---

	Introduction/Methods: I am missing a conceptual framework of quality here and it is a bit of a struggle to follow through the descriptions in the methods. The terms structure and process appear suddenly without any definitions or explanations. From the studies that you refer to in the Methods, can you develop a conceptual framework of quality in the context of delivery care – distinguishing the structural from the process indicators and also categorising them by the signal functions of routine and emergency care? A diagram would be very helpful. Methods/Results: Can you explain how the 8536 PHCs and 4810 CHCs were selected? The total number of PHCs in the country is much more (around 25,000). Were these (8536) all the PHCs for which data were available from 30 out of 36 states? But in Figure 4, only two states show complete absence of any data (not 6). This is a bit confusing. What does this mean for interpreting the results? Can you include some more explanation about this in the Methods and the Discussion. I like the way you have graded the facilities by districts in Figure 4, based on their scores. Could you bring this out in the text also. When you use words like ‘sub-optimal’ in a general way, the question that comes to my mind is, what is optimal then? 100% or 80% or 60%? A few minor comments are in the attached paper. - The reviewer also provided a marked copy with additional comments. Please contact the publisher for full details.
--	--

REVIEWER	Thomas Weiser Stanford University USA
REVIEW RETURNED	18-Dec-2017

GENERAL COMMENTS	Sharma and colleagues present their assessment of the Indian public health system’s capacity to provide peripartum and newborn care based on a representative survey of both primary and community health centers from the District Level Household and Facilities Survey. They used the results of this survey to compare infrastructural and human resources to the Indian Public Health Standards, a set of recommended standards for health facilities planning and functional care delivery. They used survey results from 8536 PHC and 4810 CHC and compared those with recommendations by the IPHS to draw their conclusions. They found relatively low numbers of deliveries for each facility (8 and 41 per month, respectively). They also found poor structure and process scores based on a facilities index score that the authors created to assess availability of resources (human, infrastructural, consumable, etc). In particular, there was little difference between PHC and CHC in terms of structure or process metrics, which is somewhat surprising given that CHCs should be better resourced and are certainly performing more deliveries overall. This is interesting study, and one that is highly relevant not only to the Indian subcontinent but also many low and middle development countries that are trying to understand and address their infrastructural weaknesses, especially vis-à-vis specific service lines such as maternal and neonatal care. I had difficulty understanding how the authors developed their score, how they derived it, and whether it truly reflects the IPHS and its application to the specific
--

	facilities. The authors also do not discuss whether these resource limitations result in differences in outcomes for patients presenting to different health facilities. Finally, I cannot tell what the actual implications are, other than to note that PHC and CHC are still under-resourced. As a side point, but one that I stumbled on a number of times, I found the use of the word “delivery care” confusing. I associate “delivery” and “care” as “care delivery” or the “delivery of care”, rather than peripartum care. I might suggest that this term be changed to make explicit that the authors are addressing peripartum care and not the delivery of care in the context of the Indian health sector. I have the following major and minor points for consideration: Major:  1. The facility capacity index is a critical component of the assessment, but I had difficulty understanding how it was created. This could be fleshed out in more detail, in particular I would suggest that a figure identifying each of the components and how they were scored would be useful. I assume it was built based on the details incorporated into table 1 – perhaps an additional column noting how each element contributed to the index, and how each input was categorized (structural along with its subdomain, and process). 2. I found the message somewhat confusing. The authors note that the facilities scored relatively poorly on the indices, but they also noted that many facilities (particularly PHC) performed relatively low numbers of deliveries (average of 8). It seems to me that facilities with low volumes are naturally going to have poor resources. So what is the message that the authors want to convey- that all facilities should be well resourced, even if they do not perform much service for which they are resourced? That seems like a poor use of resources for a country like India. Thus perhaps the argument is to create more regionalization of care for births, and encourage births to take place in the CHCs where there is higher volume. That way resources and training can be focused here given how overall poorly even the CHCs perform on the index scoring. Minor:  1. I found the introduction overly vague – it did not set up the overall point and foundational basis of the study. I would recommend revising the intro to focus on the background and why the methods and the word done in this investigation is useful to understand the state of maternal peripartum care in India. 2. As a health services researcher only peripherally involved in maternal health, the point I make about “delivery care” being expressed as “peripartum care” might help address my confusion during the review process. 3. “Data” is plural: i.e. “data are...” (see for example page 5 line 27)
--	--

VERSION 1 – AUTHOR RESPONSE

Reviewers’ comments:

Reviewer 1:

1. Page 6, second para of the Introduction: The statements (lines 29-32 and 46-51) about JSY and delivery outcomes need to be presented with more factual accuracy and in a more nuanced way. Lim’s study found an effect on perinatal and neonatal deaths which Powell Jackson’s study did not. So one could consider this as mixed evidence but not as ‘no improvement in delivery outcomes’. In

terms of MMR, none of the studies found strong associations between JSY and maternal mortality but all authors have stated that insufficient study power and large standard errors could be one of the limitations. So, one could say that although maternal mortality has declined in India (from...to ...) during this period (as SRS data show), studies have not been able to establish a strong association between JSY and mortality reductions.

Response:

Thank you for this suggestion. In the revised manuscript, we have updated the second paragraph in the introduction section to reflect the nuance you pointed out. The paragraph now reads:

In India, increased coverage of facility-based births has not successfully translated to desired improvement in health outcomes for mothers and newborns. [...]Following the launch of NHRM in 2005, institutional deliveries in rural areas have more than doubled [8], and a declining trend in maternal and newborn mortality has been noted, although strong causal evidence linking NRHM efforts to improved health outcomes for mothers and newborns is lacking [9-11]. Annual decline in neonatal mortality between 2005 and 2015 was faster than in the preceding years; however, the rate of decline is not sufficient to meet the 2030 SDG targets [12]. Inadequate quality of care, including insufficient facility readiness, and low provider skill and clinical management capacity, as evidence from low and middle income (LMIC) countries indicates, may explain why increased utilization alone may not have resulted in the desired reduction in adverse intrapartum outcomes [13, 14].

2. You are also making a very strong case in the introduction, for quality being a weak link for maternal mortality improvements, and it's okay to emphasise this in the intro since you are building a rationale for the study objectives. However, in the discussion I feel you might need to raise some issues about the relationship between and mortality outcomes also, since a recent trial of a very big quality improvement programme in UP (Better Birth safety checklist) found No difference in neonatal or maternal outcomes (see Katherine Semrau et al, NEJM 2017). So there could be other factors like poor referral management for emergencies (as Powell-Jackson suggests) or perhaps something like deliveries being conducted by untrained workers even where trained staff are available (Gaurav Sharma et al, WHO Bulletin, 2016) etc., affecting mortality outcomes, besides quality. I would also argue for the importance of good quality care from a human rights and UHC perspective, and not just for mortality reductions.

Response:

Thank you for this suggestion. We agree that improving quality of care offered to women and newborns should be motivated from a human rights and UHC perspectives. We emphasize this in the first paragraph in the introduction:

An emerging body of literature suggests that getting women to facilities alone will not suffice: what happens when a woman reaches a facility – including her right to quality antenatal, intrapartum, and postpartum care – matters [3, 4], especially given the global mandate of universal health coverage [5, 6].

We have also expanded the discussion of the complexities of linking quality to health outcomes and the need to measure quality to understand whether the health system delivers the care women and children deserve. The lines 6-17 on page 21 read:

[...] expanding data collection to capture the nature and content of care being provided is important for understanding quality of care [52]. Quality of care is complex and multifaceted. While the inputs to care measured here are a critical foundation for high-quality care, they are a poor proxy of care as delivered [53]. Even measures of process quality may be insufficient to predict quality-sensitive outcomes such as morbidity and mortality [10, 54, 55], particularly for childbirth where averting severe

outcomes requires timely recognition of complications by qualified personnel and functional referral systems to ensure a rapid response [10, 55].

Future assessments should better capture the context of care and whether women's rights to quality care are being met, with a focus on clinical effectiveness of care, patient experience, respectful care, and outcomes [32].

3. Introduction/Methods: I am missing a conceptual framework of quality here and it is a bit of a struggle to follow through the descriptions in the methods. The terms structure and process appear suddenly without any definitions or explanations. From the studies that you refer to in the Methods, can you develop a conceptual framework of quality in the context of delivery care – distinguishing the structural from the process indicators and also categorising them by the signal functions of routine and emergency care? A diagram would be very helpful.

Response:

Thank you for this suggestion. We have expanded the discussion on conceptual frameworks and added description/definitions for structural and process capacity elements. We refrained from describing a new conceptual framework or adding a diagram as this has been done already in the context of care for pregnant women and newborns in the WHO quality of care framework (Tuncalp et al. 2016, BJOG; PMID: 25929823), which we reference in this manuscript. The revised paragraph (page 10, lines 13-20) now reads:

Availability of essential physical and human resources and services is foundational to the provision of high-quality care [32]. The study objective was to evaluate basic intrapartum care capacity that should be present in all childbirth facilities. As described in the WHO quality of care framework for maternal and newborn health [32], which builds on Donebedian's framework of healthcare quality [33], structural capacity elements, such as availability of drugs and supplies, refer to necessary, but not sufficient, human and physical resource conditions for care delivery. Process capacity elements refer to facility's ability to offer evidence-based practices for routine care and management of complications, such as assisted vaginal delivery or administration of parenteral antibiotics.

4. Can you explain how the 8536 PHCs and 4810 CHCs were selected? The total number of PHCs in the country is much more (around 25,000). Were these (8536) all the PHCs for which data were available from 30 out of 36 states? But in Figure 4, only two states show complete absence of any data (not 6). This is a bit confusing. What does this mean for interpreting the results? Can you include some more explanation about this in the Methods/Results and the Discussion.

Response:

In the DLHS survey, a sample of PHCs, representative at district level, and a census of CHCs were included. This is indicated in the second paragraph of the methods section ((page 10, lines 6-7).

We note in the results section (first paragraph) that data were publicly available for only 30 out of 36 states/union territories (UTs). The six states/UTs excluded in our analysis were: (1) Gujarat, and (2) Jammu and Kashmir, and the territories (3) Dadar and Nagar Haveli, (4) Daman Diu, the (5) National Capital Territory of Delhi, and (6) Lakshadweep. The four excluded UTs are not visible in the map (Figure 4) due to their small size.

We have also added the following sentence in the discussion section (page 20, lines 21-22):
Finally, these findings may not be representative of the six states and union territories without the survey data.

5. I like the way you have graded the facilities by districts in Figure 4, based on their scores. Could you bring this out in the text also. When you use words like 'sub-optimal' in a general way, the question that comes to my mind is, what is optimal then? 100% or 80% or 60%?

Response:

Thank you for pointing this out. We have replaced "suboptimal" with the following sentences:

Both PHCs and CHCs failed to meet the national standards for basic intrapartum care capacity. (page 2, lines 19-20 [abstract])

Overall capacity for basic intrapartum care was lower than the basic IPHS standard in both PHCs (mean 0.63, standard deviation [SD] 0.23) and CHCs (mean 0.75, SD 0.17) included in this analysis. (page 15 lines 1-3 [results])

We have refrained from using a numeric cutoff in the text as IPHS standards suggest that all these requirements should be in place and there is no further guidance for thresholds.

The grading of facilities as shown in Figure 4 is described in the text in page 16, lines 1-5 in the revised manuscript.

Reviewer: 2

1. I had difficulty understanding how the authors developed their score, how they derived it, and whether it truly reflects the IPHS and its application to the specific facilities. The authors also do not discuss whether these resource limitations result in differences in outcomes for patients presenting to different health facilities. Finally, I cannot tell what the actual implications are, other than to note that PHC and CHC are still under-resourced.

Response:

We appreciate these comments and have worked to clarify the methods and implications of this work. Please see specific responses below on these points. In general, we have aimed to: make explicit the connection between the IPHS standard and the facility capacity index, contextualize the measurement of facility capacity in terms of quality measurement and health outcomes more broadly, and strengthen the discussion of policy implications such as regionalization of childbirth services.

2. The facility capacity index is a critical component of the assessment, but I had difficulty understanding how it was created. This could be fleshed out in more detail, in particular I would suggest that a figure identifying each of the components and how they were scored would be useful. I assume it was built based on the details incorporated into table 1 – perhaps an additional column noting how each element contributed to the index, and how each input was categorized (structural along with its subdomain, and process).

Response:

Thank you for this suggestion. We have expanded the description of the index creation in the Methods section (page 10 paragraph 2, within the overall description of the index from page 9 to page 12) and we have added a column in Table 1 in the revised manuscript to indicate how each element in the IPHS standard/DLHS data contributed to the index.

3. I found the message somewhat confusing. The authors note that the facilities scored relatively poorly on the indices, but they also noted that many facilities (particularly PHC) performed relatively

low numbers of deliveries (average of 8). It seems to me that facilities with low volumes are naturally going to have poor resources. So what is the message that the authors want to convey- that all facilities should be well resourced, even if they do not perform much service for which they are resourced? That seems like a poor use of resources for a country like India. Thus perhaps the argument is to create more regionalization of care for births, and encourage births to take place in the CHCs where there is higher volume. That way resources and training can be focused here given how overall poorly even the CHCs perform on the index scoring.

Response:

Thank you for this comment. We are happy to expand on the argument for regionalization of services to clarify this important implication of this work and other global research. We have revised the discussion to consider the possibility of regionalization in depth (page 18, third paragraph onwards), which reads:

Our findings indicate very low monthly delivery volume in PHCs. Bypassing of care at lower-level facilities may be driven by a myriad of reasons [42], including, but not limited to, women's preference for higher quality care when it is available, as seen in Tanzania [43]. Regardless of the reason, underutilization of PHCs represents inefficiency in the Indian health system, considering the amount of resources spent at this level of healthcare [44]. This merits attention for effective planning and resource allocation as healthcare delivery structure being developed for urban India under the National Urban Health Mission follows the same model as in rural areas [22].

A demand for quality care, regardless of the level of facility, has been noted in India. Preference for higher-level facilities with better quality is observed in the state of Kerala in India [17], whereas in Tamil Nadu, women prefer to delivery in PHCs with qualified staff providing respectful care [45]. Most high-income and many middle-income countries encourage delivery in hospitals where surgical and newborn care services are available to all women [46]. Organizing health system services around the capacity to deliver high-quality care may thus respond to women's observed preferences and improve health system performance. The breadth of quality deficits in CHCs and especially PHCs, which are under-resourced despite concerted efforts and investment of the NHM, is concerning. Rather than attempting to bring many thousand facilities up to standards, regionalization of obstetric care and quality improvement in high-volume facilities may offer a more efficient resource allocation strategy for the general population. Building on the success of NRHM in using supply- and demand-side strategies to reduce barriers to facility-based delivery [47], India may be particularly well positioned to assess the feasibility of regionalizing care. Assessment of regionalization should include checks against over-medicalization of childbirth care, which has been seen in some states [48]. That said, it is essential to strengthen lower-level facilities and improve referral linkages with higher-level facilities in areas with highly remote, rural and/or vulnerable populations, to ensure continued access to services for all, especially considering evidence from India indicating preference for PHCs when they are adequate [45, 49].

We also emphasize this important point in the conclusion (page 22, lines 13-17):

Research on regionalization is a priority as this may offer an innovative approach to ensuring quality services for mothers and newborns. The impact of regionalization strategies on facility overcrowding, performance incentive structures for frontline workers, as well as equity in service access should be important considerations of such research.

4. I found the introduction overly vague – it did not set up the overall point and foundational basis of the study. I would recommend revising the intro to focus on the background and why the methods and the work done in this investigation is useful to understand the state of maternal peripartum care in India.

Response:

Thank you for your suggestion, we have reworked the background section in the revised manuscript to improve the overall flow of the section and foundational basis of the study (pages 6-9). More specifically, we have revised the last paragraph in the introduction as follows:

The aim of this work is to understand the capacity of the Indian public health system to provide quality intrapartum care. This is an important element of understanding the effects of maternal and newborn health policies to date and setting priorities for health system strengthening going forward. In this paper, using the updated IPHS (2012) as the minimum standard for basic facility capacity for intrapartum care, we (1) assess the performance of PHCs and CHCs in India against this standard; and (2) describe differences in intrapartum care capacity between urban and rural areas, and across states.

5. As a side point, but one that I stumbled on a number of times, I found the use of the word “delivery care” confusing. I associate “delivery” and “care” as “care delivery” or the “delivery of care”, rather than peripartum care. I might suggest that this term be changed to make explicit that the authors are addressing peripartum care and not the delivery of care in the context of the Indian health sector. As a health services researcher only peripherally involved in maternal health, the point I make about “delivery care” being expressed as “peripartum care” might help address my confusion during the review process.

Response:

Thank you for drawing our attention to this. In the revised manuscript, we have changed delivery care to intrapartum care to avoid confusion.

6. “Data” is plural: i.e. “data are...” (see for example page 5 line 27; page 15 line 39)

Response:

Thank you for drawing our attention to this. We have made necessary changes in the revised manuscript.

VERSION 2 – REVIEW

REVIEWER	Thomas Weiser Stanford University, USA
REVIEW RETURNED	21-Feb-2018
GENERAL COMMENTS	The authors have adequately addressed my original concerns, and this is a nice body of work.
REVIEWER	Dr. Meenakshi Gautham London School of Hygiene and Tropical Medicine, UK (based in India)
REVIEW RETURNED	12-Mar-2018
GENERAL COMMENTS	The reviewer provided a marked copy with additional comments. Please contact the publisher for full details.